# Language and beliefs in relation to noma: a qualitative study, northwest Nigeria

Elise Farley[1,2]*, Annick Lenglet[3,4], Aisha Abubakar[1], Karla Bil[3], Adolphe Fotso[1], Bukola Oluyide[1], Simba Tirima[1], Ushma Mehta[5], Beverley Stringer[6]

**1** Médecins Sans Frontières, Nigeria, **2** Department of Public Health Medicine, University of Cape Town, Cape Town, South Africa, **3** Médecins Sans Frontières, Amsterdam, The Netherlands, **4** Department of Medical Microbiology, Radboud University Medical Center, The Netherlands, **5** Centre for Infectious Disease Epidemiology and Research, School of Public Health and Family Medicine, University of Cape Town, **6** Médecins Sans Frontières, London, United Kingdom

* noma-research@oca.msf.org

**Data Availability Statement:** MSF has a managed access system for data sharing that respects MSF's legal and ethical obligations to its patients to collect, manage and protect their data

## Abstract

### Background

Noma is an orofacial gangrene that rapidly disintegrates the tissues of the face. Little is known about noma, as most patients live in underserved and inaccessible regions. We aimed to assess the descriptive language used and beliefs around noma, at the Noma Children's Hospital in Sokoto, Nigeria. Findings will be used to inform prevention programs.

### Methods

Five focus group discussions (FGD) were held with caretakers of patients with noma who were admitted to the hospital at the time of interview, and 12 in-depth interviews (IDI) were held with staff at the hospital. Topic guides used for interviews were adapted to encourage the natural flow of conversation. Emergent codes, patterns and themes were deciphered from the data derived from IDI's and FGDs.

### Results

Our study uncovered two main themes: names, descriptions and explanations for the disease, and risks and consequences of noma. Naming of the disease differed between caretakers and heath care workers. The general names used for noma illustrate the beliefs and social system used to explain the disease. Beliefs were varied; participant responses demonstrate a wide range of understanding of the disease and its causes. Difficulty in accessing care for patients with noma was evident and the findings suggest a variety of actions taking place before reaching a health center or health worker. Patient caretakers mentioned that barriers to care included a lack of knowledge regarding this medical condition, as well as a lack of trust in seeking medical care. Participants in our study spoke of the mental health strain the disease placed on them, particularly due to the stigma that is associated with noma.

responsibility. Ethical risks include, but are not limited to the nature of MSF operations and target populations being such that data collected often involves highly sensitive data. The dataset supporting the conclusions of this article is available are available on request in accordance with MSF's data sharing policy (available at: http://fieldresearch.msf.org/msf/handle/10144/306501). Requests for access to data should be made to data.sharing@msf.org.

**Funding:** The authors received no specific funding for this work.

**Competing interests:** The authors have declared that no competing interests exist.

## Conclusions

Caretaker and practitioner perspectives enhance our understanding of the disease in this context and can be used to improve treatment and prevention programs, and to better understand barriers to accessing health care. Differences in disease naming illustrate the difference in beliefs about the disease. This has an impact on health seeking behaviours, which for noma cases has important ramifications on outcomes, due to the rapid progression of the disease.

## Author summary

Noma (cancrum oris) is an orofacial gangrene that rapidly disintegrates the hard and soft tissues of the face. Little is known about noma as most cases live in underserved and inaccessible regions. We aimed to assess the language used and beliefs around noma, in northwest Nigeria. Findings will be used to inform prevention programs. Five focus group discussions were held with caretakers of patients with noma admitted to the hospital at the time of interview, and 12 in-depth interviews were held with staff at the hospital. Our study uncovered two main themes: names; descriptions and explanations for the disease, and risks and consequences of noma. Naming of the disease differed between caretakers and heath care workers. Difficulty in accessing care for patients with noma was evident. Barriers to care and lack of knowledge and trust were evident. The impact of noma was not limited to physical presentation; stigmatisation was mentioned as a key difficulty. Differences in disease naming illustrate the difference in beliefs and has an impact on health seeking behaviour, which for noma cases, has severe ramifications due to the rapid progression of the disease.

## Introduction

Noma, also known as cancrum oris, is a neglected disease of extreme poverty. It presents as a rapidly progressing gangrenous infection of the oral cavity [1], and mostly affects children aged between two and five years[2]. It has been estimated that without treatment, up to 90% of patients with noma die within two weeks from the onset of symptoms [3], and those who survive have severe facial disfigurements[2]. Treatment in the early stages of disease with antibiotics, wound debridement and nutritional support greatly reduces mortality and morbidity [1,3]. Due to the widespread destruction of the facial structures including the cheek, nose, lips and eyes, patients with noma have multiple physical impairments such as difficulty eating, seeing and breathing, and many suffer from stigmatization in their communities[4]. Noma is thought to be most prevalent in low socio-economic regions in Africa and Asia[4]. The WHO estimate that 140,000 new cases of noma occur annually[5], however this figure is debated due to a lack of robust evidence on the epidemiology of the disease.

Since 2014, Médecins Sans Frontières (MSF) has collaborated with the Nigerian Ministry of Health (MOH) to treat patients with noma identified across the northwest of Nigeria, at the Noma Children's Hospital in Sokoto. This programme provides nutritional, psychosocial, and surgical interventions for patients with noma. Identification of patients with noma relies on active case detection within villages through extensive outreach activities and widespread communication campaigns to raise awareness of the existence of the Noma Children's Hospital.

There is still much to learn about noma as most patients live in difficult to reach areas and the disease often goes undiagnosed and is underreported. Given that many cases occur in underserved areas, few studies have aimed to explore and describe societal and community perceptions of this disease. One study in the literature examined the language used to describe noma in Hausa (the predominant language in northwest Nigeria). A term commonly used for noma was ciwon iska, which translates to 'the disease of the wind'[6]. Language used to describe diseases carry culturally determined associative meanings [7] and have been reported to affect people's conceptions of disease and the health care options they choose[8].

Understanding and using appropriate language is one of the cornerstones of ensuring an effective communication strategy with patients, as well as their families and communities. We conducted this qualitative study to gain an understanding of the locally used descriptive language and concepts of noma. Specifically, we aimed to understand the perspectives of family members of patients with noma and treating practitioners. We anticipated that our findings would inform future interventions and prevention programs.

## Methods

### Ethics

The MSF Ethics Review Board (1710), Usmanu Danfodiyo University Teaching Hospital Health Research and Ethics Committee in Nigeria (UDUTH/HREC/2017/No.595) and the Ministry of Health in both Sokoto (SKHREC/032/017) and Kebbi (MOH/SUB/4027/Vol.I/14) states approved the study protocol. Informed consent for interviews and audio recordings were sought using an information sheet translated into Hausa stating the purpose of the study and the voluntary nature of participation. All interviewees were over the age of 18 and each participant provided written informed consent (for participants who were illiterate, the consent form was read aloud to them and a thumbprint was then requested). All participants were assured that there was limited risk of harm from participation in this study, and that they were free to withdraw at any point.

### Setting

The Noma Children's Hospital in Sokoto, northwest Nigeria, has provided treatment for patients with noma for many years, and, since 2014, MSF has supported noma initiatives at the hospital.

### Recruitment and sampling

Five focus group discussions (FGDs) with adult caretakers (predominantly the mothers, grandmothers or fathers of the patient with noma) who were looking after the patients at the hospital and twelve in-depth interviews (IDIs) with healthcare staff were held in June and July 2017. Convenience sampling was paired with multivariate sampling to ensure a wide variety of participants for the five FGD's. The research assistant (AA) selected men and women for the groups separately, due to the social norms of this region, respecting any cultural sensitivities. There were three female groups and two male groups as the majority of caretakers were female. FGD's were composed of not more than eight participants at a time. Vignettes were used to encourage participant reflections [9] on relevant life memories, and current experiences related to noma were explored. Purposeful multivariate sampling [10] was used to recruit healthcare staff members for in-depth interviews in order to ensure rich descriptive data from a diverse group. Twelve staff members at the Noma Children's Hospital were selected by which time data saturation occurred[11]. Three male program management staff, one male

and four female medical team members and three male and one female paramedical staff members were selected (laboratory, mental health, pharmacy, nutrition staff).

## Research design

A descriptive qualitative research design was used in which data was gathered from caretakers of patients with noma and staff at the hospital using FGDs and IDIs guided by topic-led questions (see S1 and S2 Texts). Both IDIs and FGDs were carried out using themes relevant to the study aims, adhering to the open ended, qualitative interview procedure. The choice of methods was used to gain a rich understanding of the topic, specifically the patient and practitioner perspectives of the disease[12].

## Data collection

Interviews were audio-recorded in quiet, private, locations that were familiar to participants. All FGDs were conducted by AA in Hausa; the Principal Investigator (EF) conducted the IDIs in English. AA transcribed all interviews. AA translated transcriptions of recorded FGDs verbatim from Hausa to English. Confidentiality was enabled for all participants by replacing the names of the respondents and all data referring to them with numerical codes indicating the type of sample (IDI for in-depth interview; FGD for focus group). During the FGD, participants agreed to keep confidential what was discussed during the group session. Both FGD and IDI respondents were reassured that data that could potentially identify a person or location was anonymised using pseudonyms that could not be traced back to them. Electronic data were password protected.

## Data validation and analysis

Data validation was conducted through continual checking throughout the IDI's and FGD's; AA and EF would repeat their understanding of what participants were saying throughout the interviews to ensure a correct and clear interpretation.

Data analysis started from the moment data were gathered. Data were initially managed through reading and rereading all transcriptions of recorded conversations allowing for familiarisation and initial coding of data. EF and BS (final author) manually analysed the data by highlighting words, phrases or paragraphs, which then emerged into codes that were constantly compared. Participant responses from IDI's were compared with FGD findings, with common patterns and themes identified. As well as points of agreement, divergent themes were established through this process.

**Table 1. Names used by caretakers and Noma Children's Hospital staff (medical, paramedical and program management staff) to describe noma.**

|  | Caretakers | Staff |
| --- | --- | --- |
| Danhurawa | X |  |
| Tuareg | X |  |
| Akin | X |  |
| Ciwon iska | X | X |
| Ciwon daji | X | X |
| Ciwon/ maci dan wawa | X | X |
| Zaizayar baki |  | X |
| Noma |  | X |
| Sakiya |  | X |

## Results

We present two themes that emerged during analysis: naming and explanations for noma (Tables 1 and 2) and risks and consequences of noma (Table 3). We illustrate our findings through quotes from participants.

### Naming noma

The most commonly used name among the health care workers was noma. One important finding was that:

**Table 2. Naming and explanations of noma with illustrative quotes.**

| *Names* | |
|---|---|
| "Danhurawa" that is the name we called it. Danhurawa that is the name we called our own. Everyone that comes to see it will say yes it is the one. They will say that she is suffering from it. | FGD 1, Female |
| They called "iska" and another name is "maci dan wawa" which eats the gums that is the names I know iska and maci dan wawa. Maci dan wawa eats the teeth and eats the mouth and make a hole just like this one | FGD 2, Male |
| They called the disease "Disease of Tuareg" and another name is like akin | FGD 2, Male |
| They were saying it was ciwon daji (cancer) | FGD 3, Female |
| The word noma itself in the local language it means farming. So often they mixed it up with farming. | IDI 3, Male, Paramedical |
| The only name they given to the disease either they say ciwon daji locally which is example like is disease from the bush. | IDI 7, Male, Paramedical |
| Ciwon daji! | IDI 9, Female, Medical |
| They always call it noma | IDI 11, Female, Medical |
| *Beliefs and explanations* | |
| This disease is caused by insect, even the measles they did is caused by an insect, insect from the bush. | FGD 1, Female |
| (I) am sure it is one of the traps of iska (Jinn). | FGD 2, Male |
| Measles is the cause of the disease. | FGD 3, Female |
| Some will say it is (caused by) a bird. We told you that if someone get the disease they will say that the bird has catch him, it will make your mouth to swell up until they poke it and it will burst out and destroy the mouth by falling off. It is all happening. | FGD 3, Female |
| It is God that brings the disease; God gave you the disease | FGD 3, Female |
| It was henna that is the cause. It was henna that my mother applied on me (henna beautifully designed) then she put me on her back then a witch saw me but then nobody knew that she was a witch. She then said this henna that you put on her is so beautiful! That is all she said! Then people said that my mother should quickly cover my legs. The next day an abscess came out that is the first thing that people saw. After a week then I changed and transformed to what I am today. | FGD 3, Female |
| The patients believe that it is because of evil cast or evil eye. | IDI 1, Male, Program Staff |
| I think I have heard someone saying it's like witchcraft. | IDI 3, Male, Paramedical |
| They (patients and caretakers) have a strong belief that whatever happened is from God. | IDI 3, Male, Paramedical |
| It is just evil spirit that is responsible for it. They don't believe that it is medical problem. | IDI 6, Female, Paramedical |
| They always believe it is iska that is the belief; iska they believe is just like a wind or let us say Jinns. This Jinn are something that are in between us but we cannot see them, Jinn can appear in trees and in wind in something, or that is they believe that the Jinns are the ones that came in and cause such kind of problem. | IDI 7, Male, Paramedical |

**Table 3. Risks and consequences of noma with illustrative quotes.**

| | |
|---|---|
| *Access to Health Care* | |
| After some few days we heard the news of this hospital. They said that they can treat this type of case. Then they prayed and wish us well and success. Then we were told to hurry and go there so that we can confirm that they can really treat the disease. When we arrived, there was a test being carried out and we told that it is possible. Then they show us. All praise be to Almighty Allah he is much better now. We are optimistic that it will be successful. | FGD 4, Male |
| You will be struggling to get the medicines while the sickness will be spreading. You will be running up and down looking for treatment while disease will be expanding. | FGD 4, Male |
| I can say that noma is not widely known, and this is the only hospital which is exclusively dedicated for the noma children disease. The one of its kind and then more than 300 patients have benefited from the plastic surgeries, re-constructive surgeries. | IDI 1, Male, Program Staff |
| We go to all the local government areas in Sokoto, Kebbi and Niger State. We do active case findings that mean trying to find new patients. Also, we do health promotion along the way. | IDI 1, Male, Program Staff |
| *Impact of the Disease* | |
| People run away from whatever you use, just because someone's mouth has cut they will say they will not eat with him. | FGD 2, Male |
| When you get this disease some people will be discriminating you, some will be running away from you. But you will not be very happy in life. | FGD 3, Female |
| After some time it will form water then they will poke it (using the traditional hot iron) after that then the mouth will burst out and fall off. | FGD 3, Female |
| We were collecting traditional medicines. We were given it to him. We were using traditional medicines. Woods, powders and so many different types was given to him. | FGD 4, Male |
| The issue is that it destroys and eats the gums, destroys someone's mouth, that people is destroyed also; there is no destruction that is more than this. | FGD 2, Male |
| After some days before a week she changes and transform to this. | FGD 3, Female |
| What scared us more and makes us to quickly rushed here was that it was itching him and he was scratching it then suddenly I saw his fingers going inside. The mouth area where the disease has affected it cut and falls off. | FGD 4, Male |
| The success story I would like to share is from a patient that had noma for a long time and she was operated in Lagos without success, she was operated in Ibadan without success. Until somebody saw her (a doctor who has worked at the Noma Children's Hospital before). The doctor saw her covering her face and she wrote a letter for her to come down to Sokoto. So she got surgery about two, three surgeries. She is fine and she is even an employee of this hospital. So it is a good story. | IDI 4, Female, Medical |
| He is always afraid of people because he stays in the bush. So I have to tell him that these people you are seeing they are just like your friends and your colleagues we are here to help you, we will not do anything to harm you. Before he was discharged he has plenty friends in the ward he will go to this bed he will gist (talk) he will go to that bed jumping from one ward to the other. | IDI 9, Female, Medical |

*"The word noma itself in the local language means farming. The word noma, the same spelling, the same pronouncement, so often they (patients) refer to it in the context of farming"* (IDI 3).

Several names for the disease were given by the FGD participants:

*"In our place we called it ciwon daji"* (FGD 4).

Naming of the disease differed between caretakers and heath care workers. Table 1 explores these differences, and suggests only a few overlapping terms; ciwon iska, ciwon daji and ciwon/ maci dan wawa.

## Beliefs and explanations

Names form a part of the understanding of the disease in this setting; the broad names used for noma such as ciwon daji are described in association with the beliefs and social system used to explain the disease. Caretakers shared several terms for the disease as well as beliefs about why and how the disease is caused. The three causative categories that were identified were spirits, living creatures (insects and animals), and connections with previous illness.

Spirits including Jinn and God were frequently reported:

*"(I) am sure it is one of the traps of Jinn"* (FGD2).

*"They always believe this thing happens (because) God allows it to be"* (IDI7).

Jinn was described as the world of other creatures that we cannot see. The world of Jinn is a spiritual realm all around us filled with normal people leading normal lives. There are good Jinn and bad Jinn, and Jinn can enter human bodies and cause harm:

*"Jinn are something that are between us but we cannot see them. Jinn can appear in trees and in wind in something, Jinns are the ones that came in and cause such kind of problem (referring to noma)"* (IDI 7).

Some respondents linked the disease to evil spirits or people:

*"One of the patients believe that it is because of evil cast or evil eye"* (IDI 1).

*"It is just evil spirit that is responsible for it"* IDI6).

Various animals and insects were also mentioned as the potential cause of noma:

*"This disease is caused by insect*; *insect from the bush"* (FGD 1).

Previous infections were also noted as a potential cause:

*"Measles is the cause of the disease"* (FGD 3).

Whilst beliefs are varied, the participant responses demonstrated a broad range of understandings about the disease and as such, it is thought to exist for multiple reasons.

## Risks and consequences: Access to health care

Difficulty in accessing care for patients with noma was evident in the findings, and explored and described in the analytical framework in Fig 1. Caretakers in our study stated they had taken patients to other health centers before being advised to go to the Noma Children's Hospital:

*"We took him (to) Binji Town, Binji Hospital, they diagnose him and checked him, then they said go to Noma Hospital, that is the only place you can get treatment for this and that is why we came here"* (FGD 2).

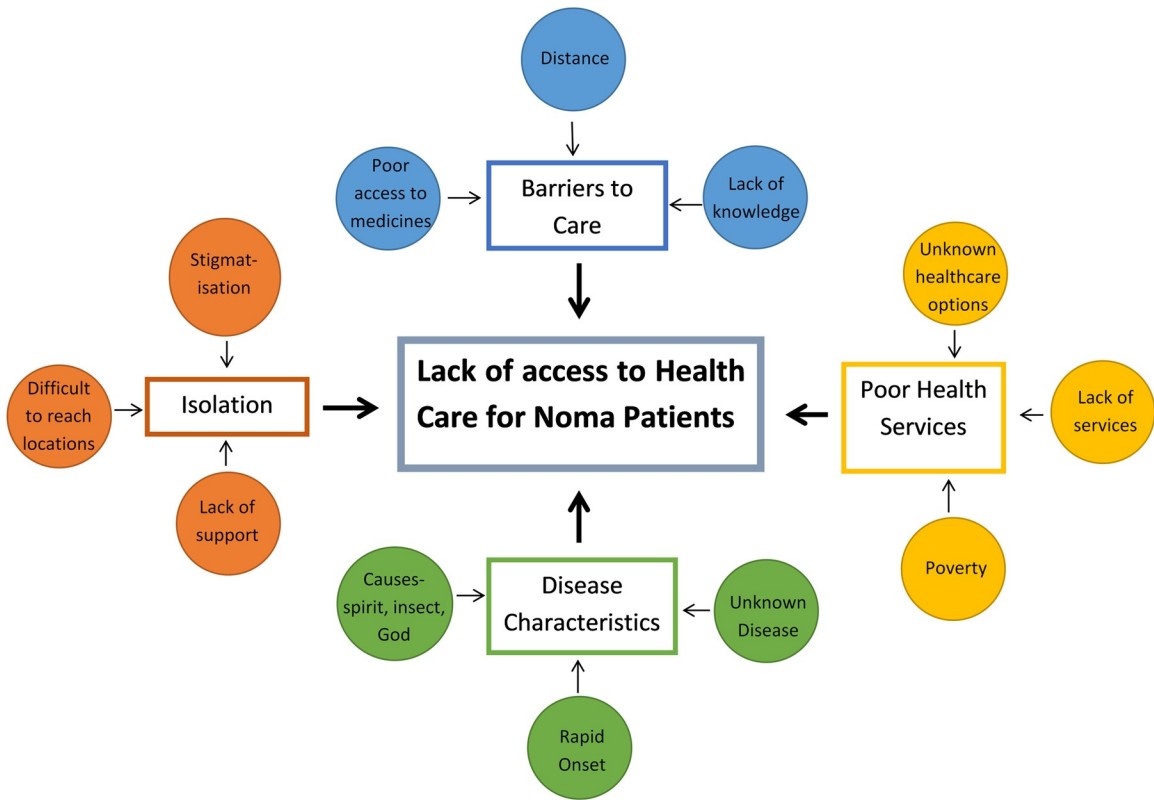

**Fig 1. Analytical framework, lack of access to care.** Reasons for the difficulties in accessing care for noma patients.

Other first points of care were traditional healers, community health centers, and private doctors. Treatment options were varied, some attempted to treat the initial oedema phase of noma:

*"After some time it will form water (swelling) then they (traditional healer) will poke it (using the traditional hot iron/ knife) after that the mouth will burst open and fall off"* (FGD 3).

Others struggled to find suitable treatment:

*"You will be struggling to get the medicines while the sickness will be spreading. You will be running up and down looking for treatment while the disease will be expanding"* (FGD 4).

This suggests a combination of actions taking place before reaching a health center or bio-medical health worker. Some participants mentioned barriers to care including a lack of knowledge about the disease and a lack of trust in the health system as well as distance from care:

*"People are coming from very far distance; they said that some people are even coming as far as Kaduna (different state) they are coming here"* (FGD 1);

Limited access to prescribed medication was also reported:

*"You will be struggling to get the medicines while the sickness will be spreading"* (FGD 4).

*"As we want to come to the hospital, everyone is saying not to go as we will just be wasting our time. Then some people said how can we see this situation of life and death and just give up, we have to try and save a life by seeking treatment?"* (FGD 1).

*"Back home they will only be trying this and that because they don't know what it is and they don't know anything about it"* (FGD 4).

## Impact of the disease

The impact of noma is not just limited to the physical presentation; participants in our study spoke of the strain on their mental health that the disease placed on them as caretakers, with stigmatisation highlighted as a key difficulty:

*"It cannot allow you to enjoy your life when you are in the company of people. You will not be happy when you are mingling with people. They will not include you in their important discussions. They will not like to sit with you. They will not eat food with you. The only thing is that you will be hiding yourself"* FGD 4.

Caretakers spoke of the physical impact of the disease and its rapid progression of the disease along with concomitant infections:

*"From the time (the patient had) measles to the time that she started this disease (noma), it was just two weeks"* (FGD 1).

The stages of noma were reported as moving rapidly from a swelling of the mouth, to the disintegration of the cheek:

*"Her own started with the swelling, then the swelling will go down and again it will swell, after some time then the mouth was destroyed on the lower lip. Then the flesh was coming out and falling off. One day the mouth fell off, it was infected and destroyed, in just a day"* (FGD 1).

Health workers spoke of destructive patient interactions and stories were mentioned which indicated the impact stigmatisation and social isolation have on the mental and physical health of patients and caretakers (also highlighted in [13–16]):

*"He (patient) is always afraid of people because he stays in the bush. So, I have to tell him that these people (other patients) you are seeing they are just like your friends and your colleagues we are here to help you, we will not do anything to harm you. Before he was discharged he has plenty friends in the ward he will go to this bed he will gist (talk) he will go to that bed jumping from one ward to the other"* (IDI 9).

The isolation that caretakers and healthcare professionals reported may influence the ability patients have to access health care and for prevention messaging to reach them. The lack of access to care for patients with noma is explored further in Fig 1.

## Discussion

### What's in a name?

Our primary emergent theme was the naming of noma in northwest Nigeria. Diseases often have multiple names, and frequently the most common name is only one of multiple reported

for a single disease. Naming of diseases can originate from visible symptoms; work on lymphatic filariasis in Nigeria has reported that local names for the disease include "elephant legs" and "swollen legs", which appropriately describe the visible manifestation of the disease[17]. Other naming options come from expected causes or, quite commonly in biomedicine, diseases are named after the individuals involved in historical descriptions, such as Alzheimer's or Parkinson's[18].

Noma is a Latinised form of a common Greek word, and is a metaphor for the continuing process of a wild fire[6]. This metaphor links conceptually with the rapid progression of the disease, as does the Hausa word ciwon iska, which loosely translates to 'the disease of the wind'[6], this could also refer to the understanding of disease transmission (spirits or animals traveling by the wind). The naming of noma as ciwon daji which loosely translates to cancer in English is also of interest, as this is similar to a biomedical name for the disease which is frequently used, cancrum oris, meaning mouth cancer[6]. Some of the other names for noma used by staff in our study were different to those used by patients. Health care workers in our study most commonly reported using the word noma when discussing the disease; this is expected, as it is the name used most commonly in the biomedical community. Noma, however is not the only name used in this community to describe this disease; other terms include necrotising ulcerative stomatitis and the aforementioned cancrum oris[6]. Our results suggest that the name most commonly used by patients or caretakers to describe noma was ciwon daji. The use of names such as ciwon iska by caretakers shows a naming system more linked to a spiritual conceptualisation of the disease, and limited biomedical understanding of the disease process.

A further novel finding from this work is that the word noma in Hausa means farming. This has the potential to cause confusion during awareness campaigns and should be taken into consideration during the planning phase. The commonly used names for identifying noma should be incorporated into all messaging used in noma prevention and treatment programs. This will help ensure clearer communication between project staff, the community, patients and their caretakers.

## Explanations for noma

The way diseases are described and understood can differ between people and groups due to a wide range of perceptions and shared social understanding of the illness, differences in language used[8], understanding of the clinical diagnosis itself[19], and the value judgements placed on these concepts[20]. Producing an explanatory model of disease [21] can provide a significant contribution to effective treatment programs and therefore positive programme outcomes. An explanatory model for noma was formed as the names used for noma were seen to be associated with a social understanding about the disease, which can be shown to impact upon what care is sought or followed by patients.

There is limited literature on the explanatory models around noma in this setting. However, a study on models for health-seeking behaviour for leprosy patients in Adamawa State, central Nigeria has shown that the majority of respondents explained the illness in terms of traditional beliefs and as 'God's wish'[22], as did patients in a further Nigerian study on orofacial clefts [23]. The leprosy and cleft participants in these studies reported seeking help from alternate health sources, for example traditional healers[22,23], as did a study assessing health seeking behaviours for hypertension in Nigeria[24]. Patients were all able to identify what disease they had by name in both the leprosy [22] and hypertension [24] studies, which indicates that these diseases are better understood in the Nigerian context than noma.

Our findings illuminated the notion that noma is linked to the spirit world, this is reiterated in a further study from northern Nigeria[6]. In that work, it was reported that not much was known about noma in those communities, and as such, broad names such as ciwon iska (disease of the wind) were used[6]. During our interviews, the caretakers explained iska as being sent by Jinn, the spirit world. This name links noma to the spirit world and offers insight into the potential explanation for noma in this community. If caretakers believe noma has a spiritual cause, this could explain why many caretakers seek care from traditional healers who typically offer treatment strategies which probe deeply into the psychological, spiritual, and social contexts of illness[25], as well as physical treatment including piercing the cheek with a sharp object and/ or offering herb mixtures to place on the wound or ingest.

Our findings suggest there is no primary explanation within this community for the disease; biomedical beliefs are less dominant, thus suggesting a pluralistic understanding of the disease. The neglected nature of this disease could contribute to and exacerbate this. As so little is known about noma globally, there is limited knowledge to share on prevention. This has an impact on the care that is sought with consequences such as poor linkage to treatment, under-reporting and poor outcomes for patients. Our findings suggest that a caretaker's knowledge and explanations for the disease may affect the health seeking treatment decisions (Fig 2). The beliefs people hold about disease have been shown in other studies to impact health seeking behaviours[26–28].

## Risk and consequences of noma

Access to health care in this part of Nigeria is difficult, especially in the rainy season, as poor infrastructure makes transportation to health facilities challenging. A lack of access to healthcare has been widely reported as a risk factor for noma development[4,29–35], and our results add weight to these assertions in that caretakers mention the difficulties they experienced accessing care for this disease. The rapid progression of noma, lack of access to care, and the delays caused by caretakers having to progress through several facilities (clinics or traditional healers), means that resulting morbidity and mortality can be severe.

The impact of noma is multifactorial; both caretakers and health workers in our study spoke of the mental health strain the disease placed on both patients and caretakers. The social isolation caused from stigmatisation of diseases is well documented[13–16]. Mental strain caused by social isolation has a wide ranging impact and can negatively affect both the mental and physical health of patients and their families[13–16]. Caretakers described the physical impact of the disease and its rapid progression along with associated links with concomitant diseases. The stages of noma were reported as moving from a swelling of the mouth, to the disintegration of the cheek, which follows clinical descriptions in the literature[31,36].

Names, beliefs, access to health care and the impact of noma are all interlinked, and form a web of issues that compound the ruthlessness of the disease. The strength of this study is that it explored the topic from two perspectives, that of the caretakers of patients, and the staff at the Noma Children's Hospital. There were several limitations including the caretakers being interviewed already being at the hospital. As such, they had likely had education on the disease from hospital staff prior to our interviews, this could have influenced the answers given. This bias could be mitigated by conducting a similar project with community members who do not have a family member affected by the disease. Further research needs to focus on the link with traditional healing, understanding the true burden of the disease and the pathogenic cause. This would enable efficient prevention programs to be formulated.

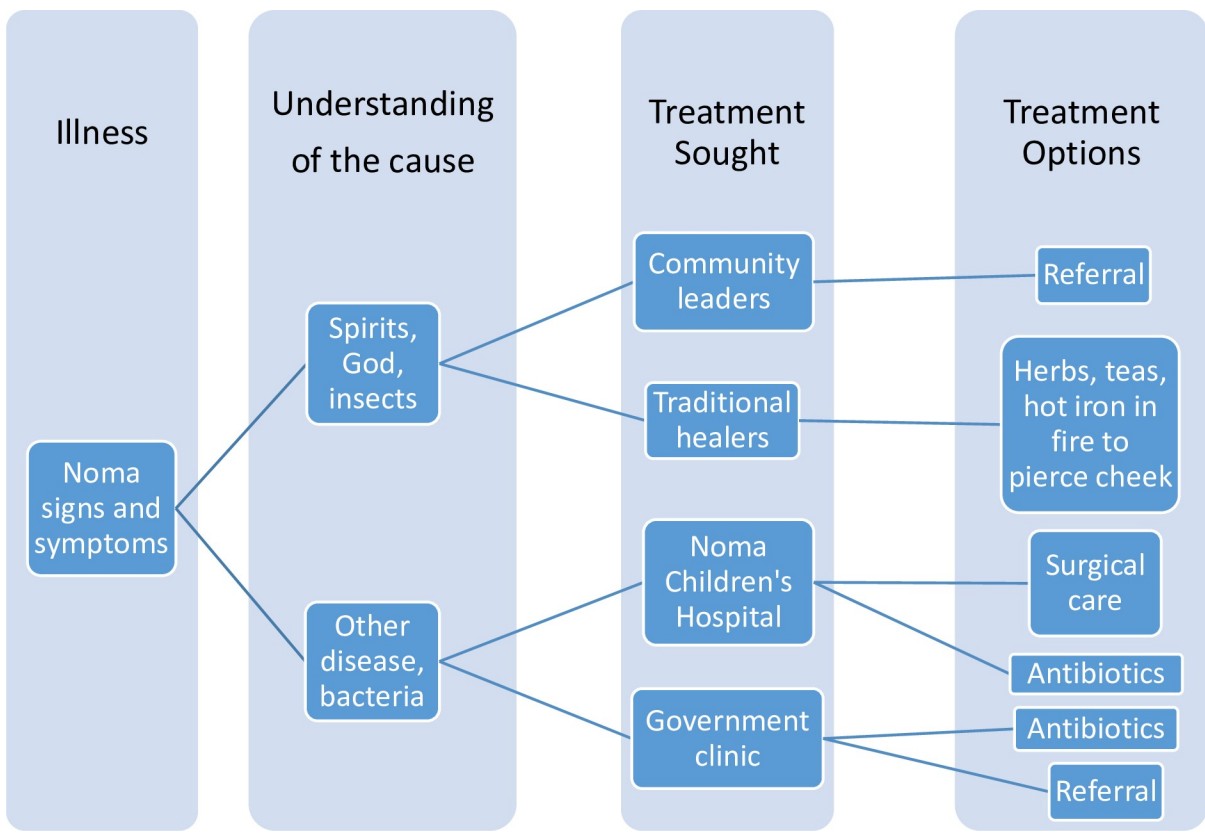

**Fig 2. Explanatory model for noma.** Knowledge and beliefs about the disease affect health seeking treatment decisions.

## Conclusions

This paper has offered an overview of the naming, explanatory models for and risks and consequences of noma in northwest Nigeria. Caretaker and practitioner perspectives enhance our understanding of the disease in this context and can be used to support case finding, referrals and addressing barriers to care. The impact of the differences noted of the names used between health workers and patients is apparent. Different naming of diseases illustrates the difference in beliefs and has an impact on health seeking behaviour, which for noma cases, has severe ramifications due to the rapid progression of the disease. Other areas where noma is endemic would benefit from similar assessments of patient caretaker and practitioner perspectives to ensure a comprehensive understanding of the contextual issues and explanatory models of the disease. The commonly used names for identifying noma should be incorporated into all messaging used in noma prevention and treatment programs.

## Supporting information

**S1 Text.**
(DOCX)

**S2 Text.**
(DOCX)

## Acknowledgments

Thank you to all the respondents in the study, for offering your time willingly. To the Noma Project team and MSF Nigerian mission for offering guidance on the local context during the conceptualisation of this project. We thank Emma Veitch (freelance editor, London) for editing assistance.

## Author Contributions

**Conceptualization:** Elise Farley, Annick Lenglet, Beverley Stringer.

**Data curation:** Elise Farley, Aisha Abubakar.

**Formal analysis:** Elise Farley, Beverley Stringer.

**Project administration:** Elise Farley, Karla Bil, Adolphe Fotso.

**Resources:** Adolphe Fotso, Bukola Oluyide.

**Supervision:** Annick Lenglet, Karla Bil, Beverley Stringer.

**Validation:** Elise Farley, Beverley Stringer.

**Visualization:** Elise Farley, Beverley Stringer.

**Writing – original draft:** Elise Farley, Beverley Stringer.

**Writing – review & editing:** Elise Farley, Annick Lenglet, Aisha Abubakar, Karla Bil, Adolphe Fotso, Bukola Oluyide, Simba Tirima, Ushma Mehta, Beverley Stringer.

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
