## [Decision Letter · Decision Letter 0]

19 Nov 2019

Dear Mrs Farley:

Thank you very much for submitting your manuscript "Language and beliefs in relation to noma: a qualitative study, northwest Nigeria" (PNTD-D-19-01408) for review by PLOS Neglected Tropical Diseases. Your manuscript was fully evaluated at the editorial level and by independent peer reviewers. The reviewers appreciated the attention to an important topic but identified some aspects of the manuscript that should be improved.

We therefore ask you to modify the manuscript according to the review recommendations before we can consider your manuscript for acceptance. Your revisions should address the specific points made by each reviewer.

(1) A letter containing a detailed list of your responses to the review comments and a description of the changes you have made in the manuscript.

(2) Two versions of the manuscript: one with either highlights or tracked changes denoting where the text has been changed (uploaded as a "Revised Article with Changes Highlighted" file); the other a clean version (uploaded as the article file).

(3) If available, a striking still image (a new image if one is available or an existing one from within your manuscript). If your manuscript is accepted for publication, this image may be featured on our website. Images should ideally be high resolution, eye-catching, single panel images; where one is available, please use 'add file' at the time of resubmission and select 'striking image' as the file type. 

Please provide a short caption, including credits, uploaded as a separate "Other" file. If your image is from someone other than yourself, please ensure that the artist has read and agreed to the terms and conditions of the Creative Commons Attribution License at http://journals.plos.org/plosntds/s/content-license (NOTE: we cannot publish copyrighted images). 

(4) Appropriate Figure Files 

Please remove all name and figure # text from your figure files upon submitting your revision. Please also take this time to check that your figures are of high resolution, which will improve both the editorial review process and help expedite your manuscript's publication should it be accepted. Please note that figures must have been originally created at 300dpi or higher. Do not manually increase the resolution of your files. For instructions on how to properly obtain high quality images, please review our Figure Guidelines, with examples at: http://journals.plos.org/plosntds/s/figures

While revising your submission, please upload your figure files to the Preflight Analysis and Conversion Engine (PACE) digital diagnostic tool, https://pacev2.apexcovantage.com/ PACE helps ensure that figures meet PLOS requirements. To use PACE, you must first register as a user. Then, login and navigate to the UPLOAD tab, where you will find detailed instructions on how to use the tool. If you encounter any issues or have any questions when using PACE, please email us at figures@plos.org.

We hope to receive your revised manuscript by Jan 18 2020 11:59PM. If you anticipate any delay in its return, we ask that you let us know the expected resubmission date by replying to this email.

To submit your revised files, please log in to https://www.editorialmanager.com/pntd/

Sincerely,

Joseph M. Vinetz

Deputy Editor

Joseph Vinetz

Deputy Editor

Reviewer's Responses to Questions

**Key Review Criteria Required for Acceptance?**

**Methods**

-Are the objectives of the study clearly articulated with a clear testable hypothesis stated?

-Is the study design appropriate to address the stated objectives?

-Is the population clearly described and appropriate for the hypothesis being tested?

-Is the sample size sufficient to ensure adequate power to address the hypothesis being tested?

-Were correct statistical analysis used to support conclusions?

-Are there concerns about ethical or regulatory requirements being met?

Reviewer #1: A limitation that the authors acknowledge, that is that caretakers interviewed in the hospital may already have gained additional knowledge regarding noma from either being in the hospital, or have a different level of understanding of noma, from caretakers who did not accompany or encourage patients to receive medical care at the hospital.

**Results**

-Does the analysis presented match the analysis plan?

-Are the results clearly and completely presented?

-Are the figures (Tables, Images) of sufficient quality for clarity?

Reviewer #1: (No Response)

**Conclusions**

-Are the conclusions supported by the data presented?

-Are the limitations of analysis clearly described?

-Do the authors discuss how these data can be helpful to advance our understanding of the topic under study?

-Is public health relevance addressed?

Reviewer #1: Dear authors, thank you for your submission.

Yes - an important limitation is discussed:

A limitation that the authors acknowledge, that is that caretakers interviewed in the hospital may already have gained additional knowledge regarding noma from either being in the hospital, or have a different level of understanding of noma, from caretakers who did not accompany or encourage patients to receive medical care at the hospital.

The conclusions are supported by the data

I see two very important findings in this manuscript:

An important finding is that patients and caretakers often use alternative names for noma, such as Ciwon Daji. Therefore, programs aimed at encouraging awareness regarding noma should consider the importance using a variety of names when communicating about noma.

A second important finding is that noma may be thought to have a spiritual cause, and this would explain why many patients first receive treatment from a traditional healer. Increased education regarding the causes of noma, and the importance of urgently receiving treatment from qualified medical personnel, may improve outcomes of noma.

I have additional recommendations regarding some changes that would improve how the manuscript reads (below)

Throughout the manuscript, would re-word 'noma patients' to 'patients with noma' 

abstract line 2 – suggest, ‘most patients live in’

line 5 – suggest ‘programs’ instead of ‘programmes’

abstract - health care center instead of centre

'Carer' was unclear to me initially when reading the manuscipt

Suggest

Patient caretaker, and would further describe who these caretakers normally are, that is, are these family members vs. other caretakers?

Suggest revising the below sentences (as revised below):

Difficulty in accessing care for patietns with noma was evident and

the findings suggest a variety of actions taking place before reaching a health center or

health worker. Patient caregivers mentioned that barriers to care included a lack of knowledge regarding this medical condition, as well as a lack of trust in seeking medical care. Participants in our study spoke of the mental health strain the disease

placed on them (remove "as carers"), particularly due to the stigma that is associated with noma.

Conclusions Paragraph: Suggest rewording to below: 

Carer and practitioner perspectives may enlighten efforts to improve outcomes (remove case finding), and to understand barriers to health care access (re-worded).

Differences in disease naming illustrates the difference in beliefs and has an impact on if and how individuals seek medical care (re-worded),

which for noma cases has important ramifications on outcomes, due to the rapid progression of the disease (re-worded).

Line 19 - Would recommend re-wording the sentence to "most patients live in"

Line 21 - suggest ‘programs’ instead of ‘programmes’

Line 35 - suggest ‘center ‘instead of ‘centre’

Line 35-37

Patient caregivers mentioned that barriers to care included a lack of knowledge, regarding the medical condition, as well as lack of trust in seeking medical care. Participants in our study spoke of the mental health strain the disease placed on them (remove "as carers"), particularly due to the stigma that is associated with noma.

Line 41 

Patient caretaker

Line 41 suggest replacing ‘enlighten’ with ‘improve’

Line 42-44

Conclusions: Carer and practitioner perspectives may enlighten efforts to improve outcomes (remove case finding), and to understand barriers to health care access (re-worded).

Differences in disease naming illustrates the difference in beliefs and has an impact on if and how individuals seek medical care (re-worded),

which for noma cases, has important ramifications on outcomes, due to the rapid progression of the disease (re-worded).

I re-worded these sentences to:

Noma (cancrum oris) is an orofacial gangrene that rapidly disintegrates the hard and soft tissues of the face. Little is known about noma, as most cases 'patients' in underserved and inaccessible regions. We aimed to assess the descriptive language used and beliefs around noma, in this region. Findings will be used to better inform prevention programs (rather than programs).

Lines 53-34

53 Five focus group discussions were held with caretakers of noma patients who were admitted to the hospital at the time of interview (re-worded)

and

in-depth interviews (suggest adding 'that') were held with staff at the

55 hospital.

Line 69 - that up to 90% of noma cases die

suggest re-wording to, that up to 90% of patients with noma die

Line 70 - suggest re-wording Noma patients to 

Patients with noma

Line 71 - difficulty eating, seeing and breathing is how the sentence is currently written;

I suppose difficulty eating can be left, but instead of seeing, what comment further, there are visual acuity issues as a result of noma, this is not obvious to me why this would be as a reader, please expound on the reason for reduced visual acuity

Noma is thought to be most prevalent in low socio-economic regions in

73 Africa and Asia[4].

‘My understanding is that the latest estimates of global burden, suggest approximately 30,000 to 40,000 cases annually, it would be helpful to comment here on the global burden of disease, regarding noma’

Line 75

again, would state to treat patients with noma, rather than to treat noma patients

Line 76 - recommend program, instead of programme

Line 78 - suggest 'within villages' instead of 'at the village level'

Line 79 - suggest adding 'noma' immediately before 'hospital'

Currently written - 80-83 -

Little is understood about noma, as most cases live in underserved, difficult to reach areas, the mortality rate is high, and the disease often goes undiagnosed and is underreported. Due to these difficulties, the perceptions of communities affected by noma have rarely been explored.

suggest re-wording as

"There is likely much to learn about noma, as most patients live in underserved and difficult to reach areas, and the disease often goes undiagnosed and is underreported. Given that many cases occur in underserved areas, few studies have aimed to explore and describe societal and community perceptions of this medical condition.

Line 87

as written - determine peoples conceptions of disease

Is 'determine' the best word choice, or perhaps a better word choice would be to

has been reported to affect how individuals perceive their condition, and the health care options they choose

Line 88 - suggest removing this sentence

Language can have a big impact on disease perceptions and health seeking behaviours.

Suggest re-wording the following paragraph (as written below)

We explored perceptions as well as the local vocabulary and expressions

94 used to describe noma in northwest Nigeria. We anticipated that our findings would then 95 inform future interventions and planning of prevention programmes.

To

As appropriate medical and descriptive language is essential to effectively communicating with patients, as well as their families and communities, we conducted this qualitative study to better understand the locally used descriptive language, and understand local concepts of noma. Specifically, we aimed to understand the perspectives of families members, regarding how these individuals view this infection. 

Line 95 - again change to programs

Line 103 - suggest either caretakers or caregivers - can you better define whether these are family members?

Line 107, recommend placing a comma after the word separately

Line 11, suggest adding a comma after 'memories'

Line 196 - would place a comma after – animals)

Line 224 – ‘centers’ instead of ‘centres’, and would place a comm after centers

**Editorial and Data Presentation Modifications?**

Reviewer #1: Throughout the manuscript, would re-word 'noma patients' to 'patients with noma' 

abstract line 2 – suggest, ‘most patients live in’

line 5 – suggest ‘programs’ instead of ‘programmes’

abstract - health care center instead of centre

'Carer' was unclear to me initially when reading the manuscipt

Suggest

Patient caretaker, and would further describe who these caretakers normally are, that is, are these family members vs. other caretakers?

Suggest revising the below sentences (as revised below):

Difficulty in accessing care for patietns with noma was evident and

the findings suggest a variety of actions taking place before reaching a health center or

health worker. Patient caregivers mentioned that barriers to care included a lack of knowledge regarding this medical condition, as well as a lack of trust in seeking medical care. Participants in our study spoke of the mental health strain the disease

placed on them (remove "as carers"), particularly due to the stigma that is associated with noma.

Conclusions Paragraph: Suggest rewording to below: 

Carer and practitioner perspectives may enlighten efforts to improve outcomes (remove case finding), and to understand barriers to health care access (re-worded).

Differences in disease naming illustrates the difference in beliefs and has an impact on if and how individuals seek medical care (re-worded),

which for noma cases has important ramifications on outcomes, due to the rapid progression of the disease (re-worded).

Line 19 - Would recommend re-wording the sentence to "most patients live in"

Line 21 - suggest ‘programs’ instead of ‘programmes’

Line 35 - suggest ‘center ‘instead of ‘centre’

Line 35-37

Patient caregivers mentioned that barriers to care included a lack of knowledge, regarding the medical condition, as well as lack of trust in seeking medical care. Participants in our study spoke of the mental health strain the disease placed on them (remove "as carers"), particularly due to the stigma that is associated with noma.

Line 41 

Patient caretaker

Line 41 suggest replacing ‘enlighten’ with ‘improve’

Line 42-44

Conclusions: Carer and practitioner perspectives may enlighten efforts to improve outcomes (remove case finding), and to understand barriers to health care access (re-worded).

Differences in disease naming illustrates the difference in beliefs and has an impact on if and how individuals seek medical care (re-worded),

which for noma cases, has important ramifications on outcomes, due to the rapid progression of the disease (re-worded).

I re-worded these sentences to:

Noma (cancrum oris) is an orofacial gangrene that rapidly disintegrates the hard and soft tissues of the face. Little is known about noma, as most cases 'patients' in underserved and inaccessible regions. We aimed to assess the descriptive language used and beliefs around noma, in this region. Findings will be used to better inform prevention programs (rather than programs).

Lines 53-34

53 Five focus group discussions were held with caretakers of noma patients who were admitted to the hospital at the time of interview (re-worded)

and

in-depth interviews (suggest adding 'that') were held with staff at the

55 hospital.

Line 69 - that up to 90% of noma cases die

suggest re-wording to, that up to 90% of patients with noma die

Line 70 - suggest re-wording Noma patients to 

Patients with noma

Line 71 - difficulty eating, seeing and breathing is how the sentence is currently written;

I suppose difficulty eating can be left, but instead of seeing, what comment further, there are visual acuity issues as a result of noma, this is not obvious to me why this would be as a reader, please expound on the reason for reduced visual acuity

Noma is thought to be most prevalent in low socio-economic regions in

73 Africa and Asia[4].

‘My understanding is that the latest estimates of global burden, suggest approximately 30,000 to 40,000 cases annually, it would be helpful to comment here on the global burden of disease, regarding noma’

Line 75

again, would state to treat patients with noma, rather than to treat noma patients

Line 76 - recommend program, instead of programme

Line 78 - suggest 'within villages' instead of 'at the village level'

Line 79 - suggest adding 'noma' immediately before 'hospital'

Currently written - 80-83 -

Little is understood about noma, as most cases live in underserved, difficult to reach areas, the mortality rate is high, and the disease often goes undiagnosed and is underreported. Due to these difficulties, the perceptions of communities affected by noma have rarely been explored.

suggest re-wording as

"There is likely much to learn about noma, as most patients live in underserved and difficult to reach areas, and the disease often goes undiagnosed and is underreported. Given that many cases occur in underserved areas, few studies have aimed to explore and describe societal and community perceptions of this medical condition.

Line 87

as written - determine peoples conceptions of disease

Is 'determine' the best word choice, or perhaps a better word choice would be to

has been reported to affect how individuals perceive their condition, and the health care options they choose

Line 88 - suggest removing this sentence

Language can have a big impact on disease perceptions and health seeking behaviours.

Suggest re-wording the following paragraph (as written below)

We explored perceptions as well as the local vocabulary and expressions

94 used to describe noma in northwest Nigeria. We anticipated that our findings would then 95 inform future interventions and planning of prevention programmes.

To

As appropriate medical and descriptive language is essential to effectively communicating with patients, as well as their families and communities, we conducted this qualitative study to better understand the locally used descriptive language, and understand local concepts of noma. Specifically, we aimed to understand the perspectives of families members, regarding how these individuals view this infection. 

Line 95 - again change to programs

Line 103 - suggest either caretakers or caregivers - can you better define whether these are family members?

Line 107, recommend placing a comma after the word separately

Line 11, suggest adding a comma after 'memories'

Line 196 - would place a comma after – animals)

Line 224 – ‘centers’ instead of ‘centres’, and would place a comm after centers

**Summary and General Comments**

Reviewer #1: (No Response)

PLOS authors have the option to publish the peer review history of their article (what does this mean?). If published, this will include your full peer review and any attached files.

Reviewer #1: No

---

## [Editor Report · Decision Letter 1]

5 Dec 2019

Dear Mrs Farley,

We are pleased to inform you that your manuscript, "Language and beliefs in relation to noma: a qualitative study, northwest Nigeria", has been editorially accepted for publication at PLOS Neglected Tropical Diseases.

Regarding your request to me regarding a fee waiver, I would like to refer you to Author Billing,  to whom you  should direct any and all fee/funding queries to authorbilling@plos.org.  Please request a waiver in an email to that address.

Before your manuscript can be formally accepted and sent to production you will need to complete our formatting changes, which you will receive in a follow up email. Please note: your manuscript will not be scheduled for publication until you have made the required changes.

IMPORTANT NOTES

* Copyediting and Author Proofs: To ensure prompt publication, your manuscript will NOT be subject to detailed copyediting and you will NOT receive a typeset proof for review. The corresponding author will have one final opportunity to correct any errors when sent the requests mentioned above. Please review this version of your manuscript for any errors.

* If you or your institution will be preparing press materials for this manuscript, please inform our press team in advance at plosntds@plos.org. If you need to know your paper's publication date for media purposes, you must coordinate with our press team, and your manuscript will remain under a strict press embargo until the publication date and time. PLOS NTDs may choose to issue a press release for your article. If there is anything that the journal should know, please get in touch.

*Now that your manuscript has been provisionally accepted, please log into EM and update your profile. Go to http://www.editorialmanager.com/pntd, log in, and click on the "Update My Information" link at the top of the page. Please update your user information to ensure an efficient production and billing process.

*Note to LaTeX users only - Our staff will ask you to upload a TEX file in addition to the PDF before the paper can be sent to typesetting, so please carefully review our Latex Guidelines [http://www.plosntds.org/static/latexGuidelines.action] in the meantime.

Best regards,

Joseph M. Vinetz

Deputy Editor

Joseph Vinetz

Deputy Editor

---

## [Editor Report · Acceptance letter]

16 Jan 2020

Dear Mrs Farley,

We are delighted to inform you that your manuscript, "Language and beliefs in relation to noma: a qualitative study, northwest Nigeria," has been formally accepted for publication in PLOS Neglected Tropical Diseases.

Best regards,

Serap Aksoy

Editor-in-Chief

Shaden Kamhawi

Editor-in-Chief
